# Improvement of Flesh Quality of Farmed Silver Carp (*Hypophthalmichthys molitrix*) by Short-Term Stocked in Natural Water

Xiaomin Miao [1], Hui Guo [1], Yong Song [1], Chunying Du [1], Jingyun Feng [1], Yixi Tao [1], Hao Xu [1,2] and Yun Li [1,2,*]

1    Fisheries and Aquaculture Biotechnology Laboratory, College of Fisheries, Southwest University, Chongqing 400715, China
2    Key Laboratory of Freshwater Fish Reproduction and Development (Ministry of Education), Key Laboratory of Aquatic Science of Chongqing, Southwest University, Chongqing 400715, China
*    Correspondence: aquatics@swu.edu.cn

**Abstract:** To investigate the effects of different raising environments on the flesh quality, the study set up three groups of silver carp (*Hypophthalmichthys molitrix*), that is, pond-farmed; short-term stocked; and ranched, grown in a natural water ranch. This study compared the differences in muscle proximate composition and amino acid composition among three groups. The results showed that there was no difference between the ranched and the stocked groups in content of crude protein, ash, total amino acids (TAA), essential amino acids (EAA), and umami-flavored amino acids (UAA), with both groups being significantly higher than the farmed group. This shows that, in terms of flesh quality, the stocked group was closer to the ranched group and better than the farmed group after being stocked in natural water for 30 days. The results suggest that the flesh quality of farmed fish was improved by short-term stocking in natural water. To elucidate the reasons of flesh quality change in the stocked group, this study compared the food composition and intestinal digestive enzyme activity in three groups, demonstrating that the ranched and stocked groups had similar food composition, with their detritus proportion lower than that of the farmed group, consuming easily digestible phytoplankton diatoms rich in amino acids and fatty acids. No significant difference was seen between the ranched group and the stocked group, for which amylase, lipase, and trypsin activities were lower than that of the farmed group. It is hypothesized that differences in environmental and food composition resulted in lower digestive enzyme activity in the ranched group and the stocked group. In summary, the short-term stocking of farmed fish in a natural water ranch can effectively improve the nutrient content, quality and flavor, and promote value of fish products.

**Keywords:** silver carp (*Hypophthalmichthys molitrix*); flesh quality; food composition; digestive enzyme activity

**Key Contribution:** This study showed that the short-term stocking of farmed fish in a natural water ranch can effectively improve the nutrient content, quality and flavor, which provides a new method for improving the quality of commercial fish, especially filtering carps such as silver carp and bighead carp; exploring new eco-farming practices; and the rationally using reservoir resources to develop ecological fisheries in freshwater ranch environments.

## 1. Introduction

Fish has always been popular because of its high protein, low fat, and easy digestibility. Currently, half of the world's fish production comes from aquaculture [1]. In China, the price of wild fish is generally higher than that of farmed fish due to the general perception among consumers that wild fish are of better quality. Flesh quality, flavor, and appearance are the main judgment criteria to evaluate the quality of fresh commercial fish [2], which has a great impact on the market price.

In order to improve the flesh quality of fish, researchers have conducted many studies. Diet, starvation, breeding environment, and season all have different degrees of influence on the nutritional index and flavor of fish muscle [3–7]. In terms of the raising environment, there have been many studies that have found that wild fish have better flavor and firmer texture compared to farmed fish, in addition to differences in muscle nutrient composition [8–10]. For example, wild sea bass (*Dicentrarchus labrax*) had lower fat content, firmer flesh, richer flavored amino acids, and more polyunsaturated fatty acids than farmed sea bass [11]. Grass carp (*Ctenopharyngodon idella*) starved for 20 d in natural lakes showed reduced odoriferous volatiles and improved muscle hardness and elasticity [12]. The above reports mainly focus on the comparison of morphology, quality, and flavor of artificially farmed and wild fish.

Silver carp (*Hypophthalmichthys molitrix*) is the main freshwater aquaculture fish in China and is referred to as one of the four major Chinese carps, together with black carp (*Mylopharyngodon piceus*), grass carp, and bighead carp (*Aristichthys nobilis*) [13]. As algae-feeding fish, the silver carp is often stocked in natural lakes in China and Southeast Asia as algae control fishes [14]. Due to its low cost and eco-friendliness, silver carp is a major freshwater aquaculture target in China [13], with Chinese silver carp accounting for 12.05% of total freshwater aquaculture production in 2021 [15]. However, significant differences in flesh quality, texture, and flavor between artificially farmed and wild silver carps have been a constraint on the quality and market value of the farmed varieties. Investigating the causes of these differences and the effects of different farming methods on the flesh quality will contribute to improvement of the quality and market value of silver carp.

Therefore, three groups of silver carp were selected as experimental materials: pond-farmed; short-term stocked; and ranched, grown in a natural water ranch located in the Three Gorges Reservoir (TGR) of the Yangtze River. This study first examined muscle proximate composition and amino acid composition of the three groups, followed by a comparison of differences in food composition and digestive enzyme activity, and explored the effects of changes in living environment on the muscle nutrient composition of algal-feeding silver carp.

## 2. Materials and Methods

### 2.1. Experimental Design

The experiment was carried out in two water environments, one is a conventional aquaculture pond (water area $2 \times 10^{-2}$ km$^2$, average water depth 2 m) (N: 29°45′, E: 106°23′) and the other is a natural water ranch (water area 6 km$^2$, average water depth 40 m) in the TGR of Zhongxian County, Chongqing, China (N: 30°19′~30°21′, E: 108°01′~108°02′). There were three experimental groups: the first group was the farmed group, in which fish had been reared in a fish pond; the second group was the ranched group grown in the TGR water ranch; and the third group was the stocked group, in which silver carp from the same pond as the farmed group were stocked in a cage (5 m × 5 m) in a water ranch for 30 d. A total of 60 silver carp individuals were used, 20 per treatment (300–350 mm in length and 550–750 g in weight). The treatment time was 26 October to 26 November, and the pH of the aquaculture pond and the water ranch were 8.1~8.3 and 8.2~8.35, respectively; the DO were 4.3~4.65 mg/L and 6.33~6.8 mg/L, respectively; and the water temperature of both varied from 18.1 °C to 21.1 °C. The basic nutritional components of the artificial compound feed used in the farmed group are shown in Table 1.

**Table 1.** The basic nutritional components of the artificial compound feed used in the farmed group (%).

| Items | Crude Protein | Crude Fat | Crude Fibre | Ash | Lys |
|---|---|---|---|---|---|
| **Content** | 31.6 | 5.0 | 8.5 | 8.2 | 1.5 |

### 2.2. Animal Ethics

Handling and care of animals were conducted based on the Guiding Principles for the Care and Use of Laboratory Animals and were approved by the Committee for Lab-

oratory Animal Experimentation at Southwest University, China on 18 July 2019 (Issue No. 2019071806).

### 2.3. Sample Collection

Fish in three treatment groups ($n$ = 20) were anesthetized with 0.1 ppm MS-222. Their body weight and length were measured, they were dissected on ice, the intestine was taken for food composition analysis and digestive enzyme activity determination, and 10 g white muscles on both sides of the spine of each fish were taken in an aseptic environment for flesh quality determination. The above samples were stored at $-80\ ^{\circ}$C for later use.

### 2.4. Experimental Methods

The determination of proximate composition was based on the method of Wu and Liu [16,17]. Moisture content was determined by drying at 105 $^{\circ}$C; crude protein content was determined by the Kjeldahl method; crude fat content was determined by Soxhlet extraction; crude ash content was determined by high-temperature cautery; amino acid content was determined using a Hitachi L-8900 amino acid analyzer.

The amino acid score (AAS), chemical score (CS), and essential amino acid index (EAAI) were calculated according to the amino acid scoring criteria (%, dry) recommended by FAO/WHO (1973) [18] and the amino acid pattern (%, dry) of egg proteins proposed by the Institute of Nutrition and Food Hygiene, Chinese Academy of Preventive Medicine [19].

The identification of intestinal contents was conducted according to the methods of Li [20]. The food mass was diluted and mixed at a ratio of 1:1 using 0.65% saline, and 0.1 mL was taken for identification and counting of the plankton species under a microscope. The identification of plankton species was conducted by referring to the atlas of freshwater microorganisms and benthos edited by Zhou et al. [21]. The weight of organisms was calculated according to the biomass criteria for each type of aquatic organism described by Zhao [22].

The activities of trypsin, lipase, and amylase were measured with relevant kits (Nanjing Jiancheng Bioengineering Institute, Nanjing, China).

### 2.5. Statistical Analysis

The data were initially processed using Excel 2019 and then statistically analyzed with SPSS 26.0 software using one-way analysis of variance (ANOVA). Duncan's multiple range test was used if a statistical evaluation comparing different group values was necessary, with $p < 0.05$ indicating a significant difference. Data were expressed as "mean $\pm$ standard deviation".

## 3. Results

### 3.1. Proximate Composition

The results of muscle proximate composition of the three groups are shown in Figure 1. The results showed that, in terms of water content, the farmed group was the highest (80.59%), followed by the stocked group (79.64%) and the ranched group (78.89%), with significant differences among the three groups ($p < 0.05$) (Figure 1A). In terms of crude protein content, the ranched group (17.66%) and the stocked group (17.61%) were significantly higher than the farmed group (16.64%) ($p < 0.05$), and there was no significant difference between the ranched group and the stocked group ($p > 0.05$) (Figure 1B). In terms of crude fat content, there was no significant difference among the three groups (FG:0.65%; RG: 0.63%; SG: 0.64%) ($p > 0.05$) (Figure 1C). In terms of crude ash content, the ranched group (1.26%) was significantly higher than the farmed group (1.13%) and the stocked group (1.15%) ($p > 0.05$), and there was no significant difference between the farmed group and the stocked group ($p > 0.05$) (Figure 1D).

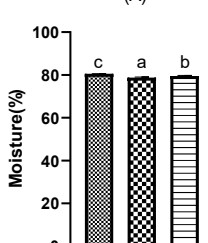 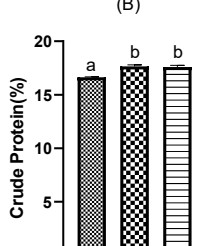 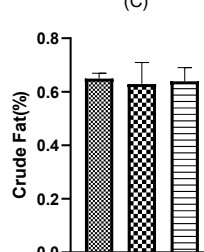 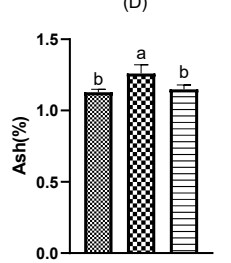

**Figure 1.** Comparison of muscle proximate composition of three groups (means ± SD, %, fresh weight) (*n* = 10). (**A**) Moisture; (**B**) crude protein; (**C**) crude fat; (**D**) ash. Different letters on the error line indicate significant differences (*p* < 0.05). The absence of statistical signs denotes no statistically significant differences among treatments (*p* > 0.05).

*3.2. Amino Acid Composition*

The amino acid composition of the three groups was similar, and the results are shown in Table 2. Seventeen common amino acids were measured in the muscle of all three groups, including eight non-essential amino acids; two semi-essential amino acids; and seven essential amino acids.

**Table 2.** Amino acid composition and contents in muscle of three groups (g/100 g, dry sample).

| Amino Acid | Farmed Group | Ranched Group | Stocked Group |
|---|---|---|---|
| Thr | 3.83 ± 0.04 [a] | 3.94 ± 0.06 [b] | 3.85 ± 0.03 [a] |
| Val | 4.43 ± 0.08 | 4.43 ± 0.05 | 4.45 ± 0.17 |
| Met | 2.74 ± 0.05 | 2.70 ± 0.05 | 2.74 ± 0.06 |
| Phe | 3.62 ± 0.05 [a] | 3.72 ± 0.04 [b] | 3.66 ± 0.11 [ab] |
| Ile | 4.16 ± 0.09 | 4.17 ± 0.06 | 4.19 ± 0.16 |
| Leu | 7.31 ± 0.08 | 7.47 ± 0.12 | 7.42 ± 0.23 |
| Lys | 8.43 ± 0.11 | 8.44 ± 0.13 | 8.43 ± 0.14 |
| EAA | 34.52 ± 0.42 [b] | 34.87 ± 0.45 [b] | 33.01 ± 0.89 [a] |
| His | 2.38 ± 0.08 [b] | 2.52 ± 0.08 [c] | 2.27 ± 0.11 [a] |
| Arg | 5.49 ± 0.08 [a] | 5.48 ± 0.08 [a] | 5.62 ± 0.15 [b] |
| HEAA | 7.87 ± 0.07 | 8.00 ± 0.03 | 7.89 ± 0.19 |
| Ser | 3.30 ± 0.09 [a] | 3.45 ± 0.05 [b] | 3.37 ± 0.06 [a] |
| Glu | 13.44 ± 0.13 [a] | 13.77 ± 0.18 [b] | 13.77 ± 0.16 [b] |
| Gly | 4.73 ± 0.06 [c] | 4.97 ± 0.08 [a] | 4.87 ± 0.78 [b] |
| Ala | 5.36 ± 0.06 [b] | 5.52 ± 0.08 [a] | 5.41 ± 0.12 [b] |
| Cys | 0.83 ± 0.05 | 0.85 ± 0.05 | 0.81 ± 0.06 |
| Tyr | 2.94 ± 0.05 [a] | 3.00 ± 0.04 [ab] | 3.01 ± 0.07 [b] |
| Pro | 3.25 ± 0.08 | 3.22 ± 0.10 | 3.19 ± 0.13 |
| Asp | 1.07 ± 0.05 | 1.06 ± 0.03 | 1.04 ± 0.05 |
| NEAA | 43.10 ± 0.44 [a] | 44.11 ± 0.73 [b] | 43.61 ± 0.90 [ab] |
| DAA | 32.80 ± 0.30 [a] | 33.59 ± 0.63 [b] | 33.22 ± 0.76 [ab] |
| TAA | 85.49 ± 0.77 | 86.99 ± 1.25 | 86.22 ± 1.82 |
| WEAA/WTAA (%) | 40.38 | 40.09 | 38.29 |
| WEAA/WNEAA (%) | 80.09 | 79.05 | 75.69 |
| WDAA/WTAA (%) | 38.37 | 38.61 | 38.53 |

Notes: Data were expressed as mean ± SD (*n* = 10). Means in a row followed by the different superscripts are significantly different (*p* < 0.05). The absence of statistical signs denotes no statistically significant differences among treatments. Abbreviations: EAA: total essential amino acids; HEAA: total semi-essential amino acids; NEAA: total non-essential amino acids; TAA: total amino acids; UAA: umami amino acids.

The Glu content was highest in all three groups, followed by Asp and Lys, while the Cys content was lowest. The total amino acid (TAA) content in the muscle of the farmed, ranched, and stocked groups was 85.49%, 86.99%, and 86.22%, respectively. Essential amino acid (EAA) content was 34.52%, 34.87%, and 33.01%, respectively. Non-essential amino acid (NEAA) content was 43.10%, 44.11%, and 43.61%, respectively. Umami amino acid (UAA) content was 32.80%, 33.59%, and 33.22%, respectively. In EAA, NEAA, and DAA,

both the ranched group and the stocked group were significantly higher than the farmed group ($p < 0.05$), with no significant differences between them ($p > 0.05$). There was no significant difference in TAA among the three groups ($p > 0.05$). The WEAA/WTAA was close to 40% and the WEAA/WNEAA was greater than 60% for all three groups.

### 3.3. Nutritional Quality Evaluation

The results of essential amino acid content and composition in the muscles of three groups compared to the FAO model and egg protein are shown in Table 3. The comparative results of amino acid score (AAS), chemistry score (CS) and essential amino acid index (EAAI) are shown in Table 4. The results show that when taking AAS as the standard, the content of Val was the lowest among the three groups, followed by Thr. That is, the first and second limiting amino acids of the three groups were Val and Thr, respectively. When taking CS as the standard, Met + Cys content was the lowest in the three groups, followed by Val. That is, the first and second limiting amino acids of the three groups were Met + Cys and Val, respectively. All three groups had AAS close to or greater than 1 and CS close to or greater than 1, and the content of Lys exceeded FAO/WHO by 1.55 times and the amino acid of egg protein by 1.91 times, respectively. The EAAI was 78.61, 79.40 and 79.09 for the farmed, ranched, and stocked groups, respectively.

**Table 3.** The essential amino acid composition and content in muscle of three groups compared to the FAO model and egg protein (mg/g. N).

| EAA | FAO Model | Egg Protein | Farmed Group | Ranched Group | Stocked Group |
|---|---|---|---|---|---|
| Thr | 250 | 292 | 239 | 246 | 241 |
| Val | 310 | 411 | 277 | 277 | 278 |
| Ile | 250 | 331 | 260 | 261 | 262 |
| Leu | 440 | 534 | 457 | 467 | 464 |
| Lys | 340 | 441 | 527 | 528 | 527 |
| Met + Cys | 220 | 386 | 223 | 222 | 222 |
| Phe + Tyr | 380 | 565 | 410 | 420 | 417 |

**Table 4.** Comparative analysis of AAS, CS and EAAI among three groups (g/100 g, dry sample).

| | EAA | Farmed Group | Ranched Group | Stocked Group |
|---|---|---|---|---|
| | Thr | 0.96 | 0.98 | 0.96 |
| | Val | 0.89 | 0.89 | 0.90 |
| | Ile | 1.04 | 1.04 | 1.05 |
| | Leu | 1.04 | 1.06 | 1.05 |
| AAS | Lys | 1.55 | 1.55 | 1.55 |
| | Met + Cys | 1.01 | 1.01 | 1.01 |
| | Phe + Tyr | 1.08 | 1.10 | 1.10 |
| | Thr | 1.31 | 1.35 | 1.32 |
| | Val | 1.08 | 1.08 | 1.08 |
| | Ile | 1.26 | 1.26 | 1.27 |
| CS | Leu | 1.37 | 1.40 | 1.39 |
| | Lys | 1.91 | 1.91 | 1.91 |
| | Met + Cys | 0.92 | 0.92 | 0.92 |
| | Phe + Tyr | 1.16 | 1.19 | 1.18 |
| EAAI | | 78.61 | 79.40 | 79.09 |

Abbreviations: AAS: amino acid score; CS: chemical score; EAAI: essential amino acid index.

### 3.4. Food Composition

The results of the food composition of the three groups are presented in Table 5, which shows that the food composition of the three groups was dominated by detritus, all above 90%, with the highest occurring in the farmed group (98.80%), followed by the stocked group (95.4%), and the ranched group (92.10%). The phytoplankton composition and average quantity of the three groups are shown in Tables 6 and 7. The results show that

diatom was dominant in the phytoplankton eaten by the ranched group and the stocked group, while cyanophyta was dominant in the farmed group.

**Table 5.** Food composition of three groups.

|  | Sediment (%) | Phytoplankton (%) | Debris (%) |
|---|---|---|---|
| **Farmed Group** | 1.05 | 0.15 | 98.8 |
| **Ranched Group** | 7.8 | 0.1 | 92.1 |
| **Stocked Group** | 4.58 | 0.02 | 95.4 |

**Table 6.** Phytoplankton composition of three groups.

| Phylum | Farmed Group (%) | Ranched Group (%) | Stocked Group (%) |
|---|---|---|---|
| Bacillariophyta | 12.0 | 78.0 | 53.1 |
| Chlorophyta | 20.0 | 2.4 | 44.1 |
| Cyanophyta | 68.0 | 19.6 | 2.7 |

**Table 7.** Average density of phytoplankton of three groups (ind.)

|  | Scientific Name | Farmed Group | Ranched Group | Stocked Group |
|---|---|---|---|---|
| Bacillariophyta |  |  |  |  |
|  | *Navicula simplex* |  | 5 | 21 |
|  | *Synedra ulna* | 1 | 7 | 20 |
|  | *Cyclotella bodanica* | 2 | 103 | 17 |
|  | *Melosira granulata (Ehr.) Ralfs.* |  | 14 | 8 |
|  | *Cymbella ehrenbergii* |  |  | 5 |
|  | *Melosira varians* |  | 2 | 5 |
|  | *Cocconeis placentula* |  |  | 1 |
|  | *Hantzschiaamphioxys* |  |  | 1 |
| Chlorophyta |  |  |  |  |
|  | *Spirogyra communis (Hass.)Kütz* |  |  | 45 |
|  | *Planktosphaeriagelatinosa* |  |  | 8 |
|  | *Chlorella vulgaris* | 5 | 4 |  |
|  | *Sceaedesmus quadricauda* |  |  | 4 |
| Cyanophyta |  |  |  |  |
|  | *Oscillatoria princeps* | 17 | 33 | 4 |

*3.5. Digestive Enzyme Activity*

The results of digestive enzyme activity for the three groups are shown in Figure 2. The trypsin, lipase, and amylase activities of the farmed group were significantly higher than those of the ranched group and the stocked group ($p < 0.05$), and the difference between the ranched group and the stocked group was not significant ($p > 0.05$) (Figure 2).

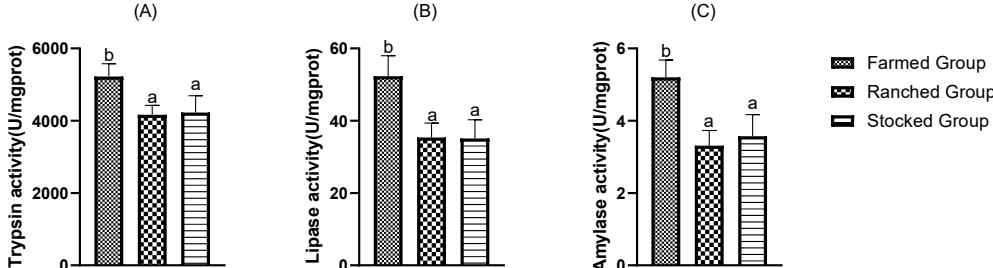

**Figure 2.** Results of digestive enzyme activities of three groups. (**A**) Trypsin activity; (**B**) lipase activity; (**C**) amylase activity. Data are expressed as mean ± SD (*n* = 10). Different letters on the error line indicate significant differences (*p* < 0.05). The absence of statistical signs denotes no statistically significant differences among treatments (*p* > 0.05).

## 4. Discussion

The results of muscle proximate composition for the three groups show that the contents of crude protein and ash were significantly higher in the ranched group and the stocked group than in the farmed group, while the differences in the content of crude fat were not significant among the three groups. The main nutrition of fish is distributed in the muscle, and the content of crude protein, fat, and ash indicate the muscle nutritional value [23]. This indicates that the nutritional value of silver carp was better in the ranched and stocked groups than in the pond-reared group, which is consistent with the results of the comparative analysis of the muscle nutritional composition of farmed and wild bass (*Dicentrarchus labrax*) [8].

In terms of essential amino acid and umami amino acid content, the stocked group was significant closer to the ranched group, both of which were significantly higher than the farmed group. The type, quantity, and composition ratio of amino acids are important indicators of the nutritional value of protein in food [24]. For example, as one of the essential amino acids, the most important physiological function of Lys is to participate in the synthesis of proteins, so it is closely related to animal growth [25]. Glu, Asp, Gly, and Ala are known as umami amino acids, and the high content of them was able to improve the quality and flavor of fish muscle [26,27]. In addition, Buchtov et al. [28] suggested that high levels of essential amino acids could also improve the flavor and quality of fish muscle. This suggests that the muscle amino acid composition was superior and the flavor was better in the ranched group and the stocked group than in the farmed group. In addition, the muscle amino acid composition of all three groups conformed to the FAO/WHO ideal pattern, with WEAA/WTAA around 40% and WEAA/WNEAA both greater than 60% [29]. AAS, CS, and EAAI are commonly used indicators for evaluating the composition of essential amino acids in fish muscle [30]. The AAS was close to or greater than 1 and the CS was greater than 0.5 in all three groups, but the EAAI was greater in the ranched group and the stocked group than in the farmed group, indicating that the muscle essential amino acid composition of the three groups was relatively balanced and rich in high-quality protein, but the ranched group and the stocked group were better than the farmed group. This is in agreement with the results for the comparative analysis of muscle nutrient composition of farmed and simulated ecologically farmed loach (*Misgurnus anguillicaudatus*) [31]. In summary, after 30 days of being stocked in a natural water ranch, the muscle quality of the farmed silver carp was improved and close to that of the wild silver carp, with a more balanced amino acid composition, richer content, and better flavor.

Silver carp is a filter-feeding fish. It feeds mainly on diatoms (Bacillariophyta) in reservoirs and rivers, while it feeds mainly on blue-green algae (Cyanophyta) in ponds [32,33]. In this study, the main diets of the ranched group and the stocked group were detritus and diatoms, while the main diet of the farmed group was detritus and blue-green algae. For fish, diatoms are more digestible than blue-green algae [34] and are more nutritious and richer in essential amino acids and polyunsaturated fatty acids [35]. Changes in the food

composition due to the different rearing waters may have contributed to the differences in flesh quality among the three groups.

Determination of digestive enzyme activities in the three groups revealed that the lipase, amylase, and trypsin activities were significantly lower in the ranched group and the stocked group than in the farmed group, and the difference between the ranched group and the stocked group was not significant. Similar results were found in both indoor and wild tuna (*Euthynnus affinis*) [36]. Digestive enzymes are mainly secreted by the digestive glands and the digestive system plays a nutritional and digestive role; therefore, changes in digestive enzymes activity can affect the growth of fish [37]. Changes in rearing water environment and food composition can affect digestive enzyme activity and cause changes in metabolism, thereby altering nutrients in the body [38,39]. This experiment conducted sampling at the end of November, when plankton was relatively unabundant in natural water and the ranched group and the stocked group were consuming less food and therefore had reduced secretion and low activity of digestive enzymes [40]. When fish ingest less food, they consume their own stored fat and glycogen to sustain life activities, while amino acids are retained as functional substances and their levels tend to increase [41]. It is therefore hypothesized that differences in environmental and food composition resulted in lower digestive enzyme activity in the ranched group and the stocked group. Both groups consumed their own fat and glycogen, in addition to ingestion and digestion, and they consumed more nutrient-rich diatoms, resulting in higher crude protein, ash, and amino acid content, which leads to higher flesh quality.

## 5. Conclusions

It has been shown that, for the same species of fish, both nutritional quality and palatability are superior to pond-cultured groups under wild and extensive water culture conditions, suggesting that we can improve the quality of fish by changing the culture environment, but the advantages of pond culture in terms of low cost and high output cannot be ignored. In this study, by stocking pond-cultured silver carp in natural waters for 30 d and comparing the changes in muscle quality of the stocked silver carp, the results showed that initially, in silver carp pond-reared through 30 days of short-term stocking in the natural ranch waters of the TGR, the basic nutrient composition and amino acid composition changed significantly, with no significant difference compared to the group grown for a longer time in ranch water, and all indexes were better than those of the farmed group. Analysis of food composition and digestive enzyme activity showed no significant differences between the ranched group and the stocked group, and significant differences for the farmed group. In summary, changes in some muscle nutrients caused by differences in environmental and food composition are responsible for the improved quality of farmed silver carp after short-term stocking in natural water. The transfer of farmed fish to natural water for a certain period can effectively improve the nutrient composition, quality, and flavor, indicating that the freshwater ranch-stocked model is a potentially viable eco-farming model, which can not only improve the muscle quality of cultured fish, but also make efficient use of reservoir resources. The present study provides a new method for improving the quality of commercial fish, especially filtering carps such as silver carp and bighead carp; exploring new eco-farming practices; and the rationally using reservoir resources to develop ecological fisheries in freshwater ranch environments.

**Author Contributions:** Conceptualization, X.M. and H.G.; methodology, X.M., H.G. and Y.S.; software, X.M., C.D. and J.F.; validation, X.M., H.G. and Y.T.; formal analysis, X.M., C.D. and J.F.; investigation, X.M., H.G. and H.X.; resources, Y.L.; data curation, X.M., H.G. and Y.S.; writing—original draft preparation, X.M., H.G., Y.S. and Y.L.; writing—review and editing, all; visualization, X.M. and Y.T.; supervision, X.M. and Y.L.; project administration, Y.L.; funding acquisition, Y.L. All authors have read and agreed to the published version of the manuscript.

**Funding:** The study was supported by the Key Research Project of Chongqing Fishery Science and Technology Innovation Alliance (no. 4322200053), the Evaluation on the Effectiveness of Fishery Stock Enhancement Project of Chongqing Municipal Agricultural and Rural Committee (CQLT-2022-011), and the Ecological Fishery Technological System of Chongqing Municipal Agricultural and Rural Committee under Grant (no. 4322000101).

**Institutional Review Board Statement:** Handling and care of animals were conducted based on the Guiding Principles for the Care and Use of Laboratory Animals and were approved by the Committee for Laboratory Animal Experimentation at Southwest University, China on 18 July 2019 (Issue No. 2019071806).

**Data Availability Statement:** Data is contained within the article. The data presented in this study are available in [insert article].

**Conflicts of Interest:** The authors declare no conflict of interest.

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
