# Peer review of "Improvement of Flesh Quality of Farmed Silver Carp (Hypophthalmichthys molitrix) by Short-Term Stocked in Natural Water"

_fishes, doi:10.3390/fishes8030142_

Round 1
Reviewer 1 Report
This study provides novelty information related to rearing methode and its effect on flesh quality. However, a major revision in the methods and result are mandatory. Pay more attention to experimental treatment design and the statistical result. Please have a look at the attached file for more suggestions.

Author Response
Dear Reviewer:
We really appreciate all the valuable comments you have provided, which are very helpful for revising and improving our manuscript “Improvement of Flesh Quality of Farmed Silver Carp (Hypophthalmichthys molitrix) by Short-term Stocked in Natural Water”. We have carefully considered your comments, and have marked revisions in red in the revised paper. The detailed reply is attached.
We would like to take this opportunity to express our gratitude to you in helping us to clarify a few points in our revised manuscript. Your constructive comments have enabled us to arrive at an improved manuscript. If there are other aspects in the manuscript that require further clarification, kindly let us know and we would be delighted to comply with.
We look forward to hearing from you soon about the suitability of this manuscript for publication in Fishes.
Thanks again for all the helpful comments you provided.
Sincerely yours,
Yun Li, Ph.D.
College of Animal Science and Technology
Southwest University
Tiansheng Road No.2
400715, Beibei, Chongqing, P.R. China
Tel: 86-23-68251962 (O)
Fax: 86-23-68251196
E-mail: aquatics@swu.edu.cn

Reviewer 2 Report
The paper “Improvement of Flesh Quality of Farmed Silver Carp 2 (Hypophthalmichthys molitrix) by Short-term Stocked in Natural Water” is clearly written and sound scientific high.
The study design sounds good and is well describe and also results are well explained; notwithstanding I think that conclusion could be written only about the carp farming system and it is difficult to project results on the whole fish farming system. Results are in relation with water system (salt water/temperature and so on.
Furthermore I think that data on quality of meat could be insert and if possible also data on panel test.
Author Response

(The authors gave the same response as above.)

Reviewer 3 Report
Dear Authors
The entitled manuscript “Improvement of Flesh Quality of Farmed Silver Carp (Hypophthalmichthys molitrix) by Short-term Stocked in Natural Water” attempts to provide basic information on improving the quality of the flesh of cultivated silver carp through proximate tissue analysis. It is a well-structured study with interesting results for farmed fish quality.
In the methodology:
Line 77-81, you mention that during the rearing period, three groups consisting of one fish pond, one cage, and the ranch group (TGR) were without giving us any information about their dimension, volume, etc. Please add to the text accordingly.
The question arises here is: why did you choose only one cage, pond, and TGR group for the breeding? it is not clear if you were using the above with their replicates. Please explain!
Line 81… the rearing period for such a study was relatively short?
Line 82 states that in the experiment was used 20 silver carp.
It would be better to rewrite the sentence: 60 silver carp individuals were used, divided by 20 per treatment…….
Line 84 . at the end of the paragraph, please give us some information about the food offered, chemical composition, etc., throughout the breading
Line 91 please rewrite as Fish were anesthetized with….
Line 93-95, please add how many fish tissue (of white muscle) samples were taken for the analysis. Moreover, the samples were taken in an aseptic environment; please mention it!
Line 110 please rewrite as …aquatic organism described by Zhao [21]
Line 114-116 add a missing reference, Furthermore
please indicate if the means of all the variables obtained were analyzed for normality and homogeneity tests. Also, in the above lines, you mention that the samples were analyzed
using One-way ANOVA, and do not mention which test was performed if significant differences were observed? .
Line 115 with P<0.05 rewrite as p<0.05
in the Results
Lines 119-186. Please rewrite P< 0.05 or P>0.05 as p<0.05 or p>0.05
Lines 151-153 rewrite as Data were expressed as mean ± SEM (n=…..). Means in a row followed by the same superscript are not significantly different (p>0.05).
Line 169 in Table 2 compares the values obtained from the results of the present experiment compared to the FEA and the eggs. Then write to the heading of the table 2.
Also, in the footnotes of tables 2, 3, 4, and 5, added: Data were expressed as mean ± SEM (n=…..). The absence of statistical signs denotes no statistically significant differences among treatments (p<0.05)
In conclusion, the question arises, how this result translates into practice? Can you explain briefly?
Lines 287-379, the DOI format according to the journal guidance is https://doi.org/..... , please revise all the references.
Author Response

(The authors gave the same response as above.)

Round 2
Reviewer 1 Report
The author has made a significant change in the revised manuscript. However, some minor issues still need to be addressed. After revision, the manuscript may be accepted for publication.

Author Response
Dear Reviewer:
We really appreciate all the valuable comments you have provided, which are very helpful for revising and improving our manuscript “Improvement of Flesh Quality of Farmed Silver Carp (Hypophthalmichthys molitrix) by Short-term Stocked in Natural Water”. We have carefully considered your comments, and have revised and explained in the revised paper. The detailed reply is attached.
We would like to take this opportunity to express our gratitude to you in helping us to clarify a few points in our revised manuscript. Your constructive comments have enabled us to arrive at an improved manuscript. If there are other aspects in the manuscript that require further clarification, kindly let us know and we would be delighted to comply with.
Thanks again for all the helpful comments you provided.
Sincerely yours,
Yun Li, Ph.D.
College of Animal Science and Technology
Southwest University
Tiansheng Road No.2
400715, Beibei, Chongqing, P.R. China
Tel: 86-23-68251962 (O)
Fax: 86-23-68251196
E-mail: aquatics@swu.edu.cn

Reviewer 3 Report
According to the revised version resubmitted by the authors, this manuscript meets the criteria for publication in the present form since the authors have considered all the points indicated by the reviewers.
Author Response
Dear Reviewer:
Thank you for affirming our article. We really appreciate all the valuable comments you have provided, which have enabled us to arrive at an improved manuscript.
Thanks again for all the helpful comments you provided.
Sincerely yours,
Yun Li, Ph.D.
College of Animal Science and Technology
Southwest University
Tiansheng Road No.2
400715, Beibei, Chongqing, P.R. China
Tel: 86-23-68251962 (O)
Fax: 86-23-68251196
E-mail: aquatics@swu.edu.cn